# Enhanced Visible-Light Photocatalysis of Nanocomposites of Copper Oxide and Single-Walled Carbon Nanotubes for the Degradation of Methylene Blue

**Kamal Prasad Sapkota [1,2], Insup Lee [1], Md. Abu Hanif [1], Md. Akherul Islam [1], Jeasmin Akter [1] and Jae Ryang Hahn [1,3,*]**

[1]   Department of Chemistry and Bioactive Material Sciences, Research Institute of Physics and Chemistry, Jeonbuk National University, Jeonju 54896, Korea; mychemistry2037@gmail.com (K.P.S.); insup@jbnu.ac.kr (I.L.); hanif4572@gmail.com (M.A.H.); akherulraju@gmail.com (M.A.I.); tina44445@gmail.com (J.A.)

[2]   Department of Chemistry, Amrit Campus, Tribhuvan University, Kathmandu 44618, Nepal

[3]   Textile Engineering, Chemistry and Science, North Carolina State University, 2401 Research Dr., Raleigh, NC 27695-8301, USA

*   Correspondence: jrhahn@jbnu.ac.kr

**Abstract:** We report enhanced catalytic action of a series of copper(II)-oxide-single-walled carbon nanotube (CuO-SWCNT) composite photocatalysts (abbreviated as CuO-SWCNT-0.5, CuO-SWCNT-2, and CuO-SWCNT-5, where 0.5, 2, and 5 represent the calcination time in hours) synthesized via recrystallization followed by calcination. The photocatalytic performance of the fabricated nanocomposites was examined by evaluating the degradation of methylene blue (MB) under irradiation with visible light. All of the as-fabricated nanocomposites were effective photocatalysts for the photodegradation of a MB solution; however, the CuO-SWCNT-5 displayed the best photocatalytic ability among the investigated catalysts, achieving 97.33% degradation of MB in 2 h under visible-light irradiation. The photocatalytic action of the nanocomposites was remarkably higher than that of pristine CuO nanocrystals fabricated using the same route. The recyclability of the photocatalyst was also investigated; the CuO-SWCNT-5 catalyst could be reused for three cycles without substantial degradation of its catalytic performance or morphology.

**Keywords:** CuO-SWCNT nanocomposites; recrystallization; methylene blue; photocatalyst; visible light photocatalysis; decomposition

## 1. Introduction

Industrial effluvia, in most cases, carry severely toxic and persistent organic wastes. Such wastes are often released directly into water resources and give rise to grievous environmental pollution. Some such contaminants are genotoxic and also pose a threat to human hormonal coordination, even at low concentrations [1]. Hence, diverse treatment mechanisms have been developed to eliminate such persistent organic pollutants from wastewater [2]. Among the various technologies, photocatalysis is considered particularly promising because of its ability to generate potent oxidizing radicals in aqueous media under solar radiation; these radicals subsequently mineralize organic pollutants completely via oxidation reactions [3]. Other advantages associated with photocatalysis include low operating costs, ease of access, remarkable performance, and proficient action even under ambient environmental conditions.

Copper(II) oxide (CuO) is a narrow-bandgap semiconductor with a bandgap energy that ranges from 1.2 to 2.6 eV depending on the fabrication conditions. It exhibits *p*-type semiconductivity because it contains oxygen vacancy defects [4–6]. CuO has been used in diverse applications such as energy storage, optoelectronics, solar photovoltaics, gas and vapor sensors, high-$T_c$ superconductors, heterogeneous catalysts, and photocatalysts [7]. Nanosized CuO has been explored as an efficient photocatalyst because of its high availability, low cost, low toxicity, high catalytic activity, contracted bandgap, excellent chemical stability, and facile synthesis [8,9]. Nonetheless, CuO alone suffers a few shortcomings as a photocatalyst. Relatively weak visible-light absorption, poor accessibility of its reaction sites, and a high recombination rate of light-induced electron–hole pairs are the main concerns [10].

Carbon nanotubes (CNTs) are used extensively in various applications because of their peculiar structure (atomically sharp tip and a high aspect ratio), superb thermal and electrical conductivities, high specific surface area, large electron-storage capacity, ultra-strong mechanical properties, and excellent chemical and thermal stability [11–13]. The walls of CNTs can easily combine chemically with semiconductor nanoparticles to form valuable composites. The chemical combination of CNTs with CuO results in a multifunctional entity with superior features [14]. Hybrid nanocomposites comprising crystalline CuO attached to the walls of CNTs are used extensively in photocatalysis, gas sensing, nonenzymatic glucose sensing, electrochemical sensing, and electrochemical energy storage [2–4,8–13,15–19].

Researchers have developed several original approaches for overcoming the drawbacks and enhancing the catalytic action of CuO. The coupling of CuO with other semiconductors has been found to substantially enhance the separation and rapid transfer of photogenerated pairs of electrons and holes, augmenting the visible-light absorption capability of CuO and increasing the availability of reaction sites [20]. Pradhan et al., for example, synthesized CuO-$Co_3O_4$ nanofibers enriched with mesoporous and interconnected nanoparticles and found that CuO is a proficient visible-light-responsive cocatalyst and photogenerated-charge-pair separator [21]. Shi et al. fabricated heterostructures using photoactive copper oxide and cobalt oxide nanowires and used them to photodegrade phenol [22]. Chen et al. reported visible-light photoactivity through interfacial transfers of electrons and holes between $CuWO_4$ and CuO [23]. Improvements in the photocatalytic efficiency of CuO through the formation of heterojunctions with single-walled carbon nanotubes (SWCNTs) or multiwalled carbon nanotubes have been reported in numerous other works [24–27].

In the present work, we report the design and preparation of CuO-SWCNT photocatalysts with excellent separation and an instant transfer propensity of photon-induced electron–hole pairs, enhanced absorption of visible light, and remarkable photocatalytic efficiency. We synthesized CuO-SWCNT nanocomposites using cost-effective, facile recrystallization and calcination methods. The nanocomposites are named CuO-SWCNT-0.5, CuO-SWCNT-2, and CuO-SWCNT-5 on the basis of the duration of the calcination process at a fixed temperature (500 °C). To examine the photocatalytic performance of the fabricated nanocomposites, we used them as photocatalysts for the mineralization of methylene blue (MB) under natural sunlight. The recycling efficiency of the photocatalysts was explored through their use for three repeated cycles. We found no notable deterioration in the photocatalytic performance or the morphology of the photocatalysts after their recycling. Furthermore, our straightforward method to synthesize effective and recyclable CuO-SWCNT photocatalysts is the novelty of our work. Our procedure comprises a facile one-pot-two-chemical method and was not found to be reported in the literature. Moreover, our product is more efficient in comparison to CuO-based photocatalysts fabricated through other methods reported in the literature.

## 2. Results and Discussion

### 2.1. Morphological Characterization of the CuO-SWCNT Photocatalysts

　　Figure 1 displays FE-SEM micrographs of the synthesized CuO-SWCNT nanocomposite. Each SWCNT is surrounded by multiple CuO nanocrystals. The red arrows indicate some of the SWCNTs in the composite (further analysis was performed by high resolution transmission electron microscopy (HR-TEM), as discussed later in this section). The CuO nanocrystals are arranged randomly around each SWCNT. More CuO nanocrystals are gathered in the areas with more SWCNTs, whereas no CuO nanocrystals are observed in the regions with no SWCNTs. The space between the consecutive groups of CuO nanocrystals demonstrates the distance between SWCNTs. The CuO nanocrystals exhibit different sizes and are randomly distributed in the space. The CuO nanocrystals are nanometer-scale; however, the particle size of the composite appears to depend on the number of aggregating nanocrystals neighboring the SWCNTs.

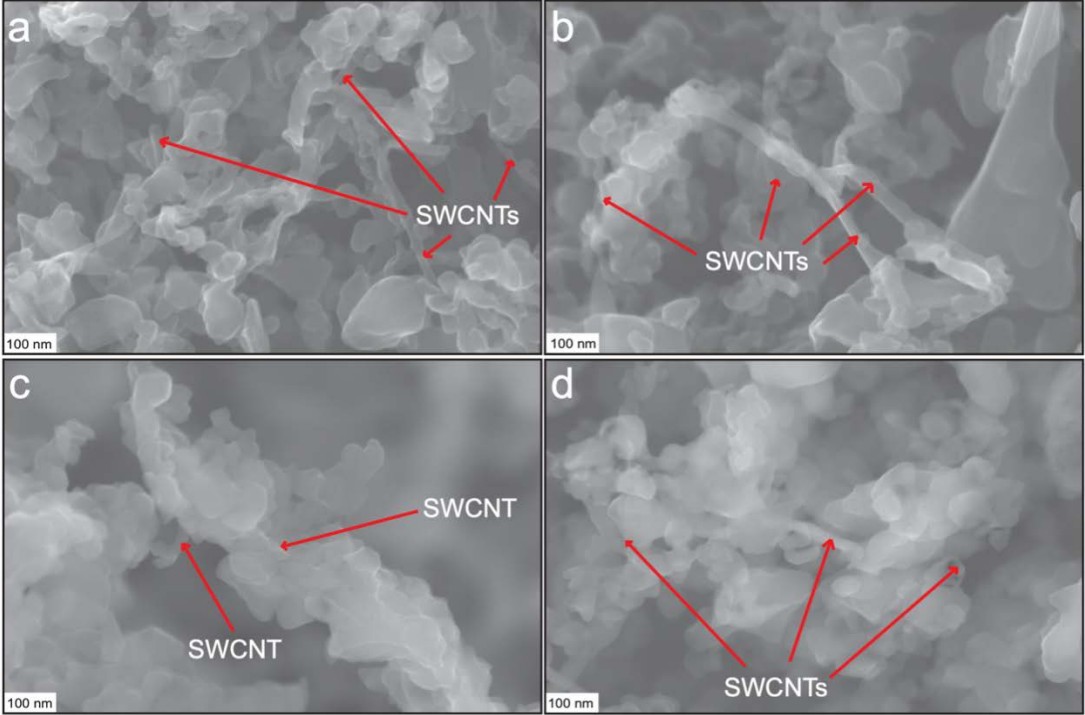

**Figure 1.** (**a**)–(**d**) FE-SEM images of copper(II)-oxide-single-walled carbon nanotube (CuO-SWCNT) nanocomposites. SWCNTs (designated by red arrows in each image) are surrounded by CuO nanocrystals in the CuO-SWCNT nanocomposites.

　　The nanostructure and the association of components in the CuO-SWCNT nanocomposite were examined via HR-TEM; the representative micrographs are presented in Figure 2. Images of nanocomposites at successively higher magnifications are displayed in Figure 2a–d. Both the components of the nanocomposite, i.e., CuO nanoparticles and SWCNTs, are conspicuous. An interconnected network of constituents is observed in the composite. The images show that CuO nanoparticles are firmly attached to the SWCNTs, forming CuO-C heterojunctions. These heterojunctions between the constituents are critical for achieving semiconductor nanocomposites with robust photocatalytic activity. The CuO nanoparticles appear to be crystals with monoclinic structures. The crystals are small and well-distributed, with sizes ranging from 4.62 to 11.15 nm.

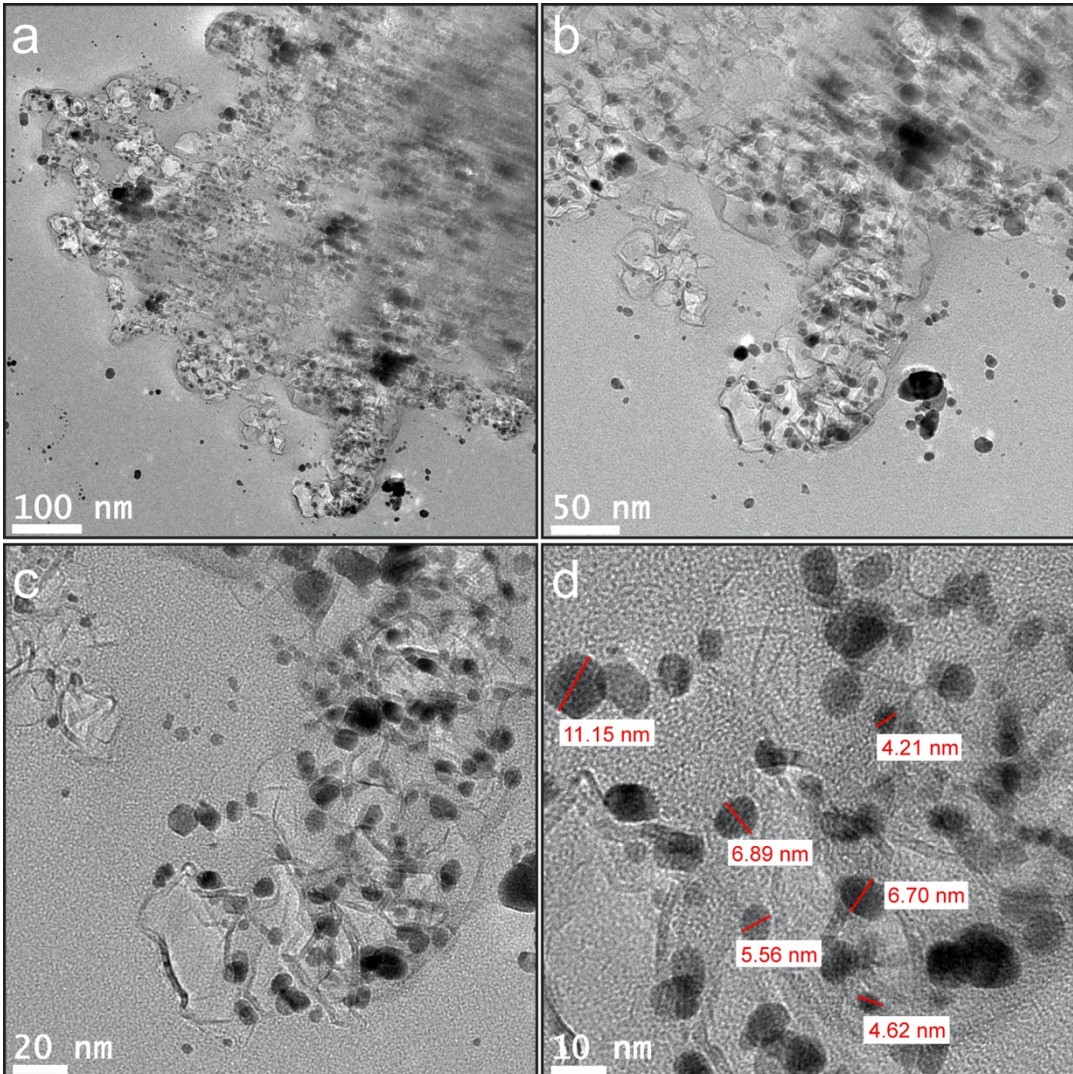

**Figure 2.** (**a**)–(**c**) HR-TEM micrographs of CuO-SWCNT nanocomposite with successively higher magnification (**d**) Measurement of particle size of CuO nanocrystals.

### 2.2. Structural Characterization of CuO-SWCNT Nanocomposites

The structural characterization of the CuO-SWCNT nanocomposites was conducted through X-ray diffractometric analysis (XRD), and the structural features of the CuO-SWCNT nanocomposites are compared with those of pure CuO and pure SWCNTs in Figure 3. The peaks at 26.03° and 43.32° in the XRD pattern of the pristine SWCNTs (curve a in Figure 3a) correspond to the (002) and (100) planes of graphite carbon, respectively. The peak at 26.03° specifies the *d*-spacing of carbon, which is characteristic to the crystalline form. The peak at 43.32° indicates the presence of some disordered carbon [1]. The characteristic diffraction peaks at 36.70°, 43.53°, 50.71°, and 74.35° in the pattern of the pristine CuO (curve b in Figure 3a) correspond to the (002), (111), (202), and (222) planes of CuO, respectively. The observed peaks confirm that the structure of the CuO phase is monoclinic [3].

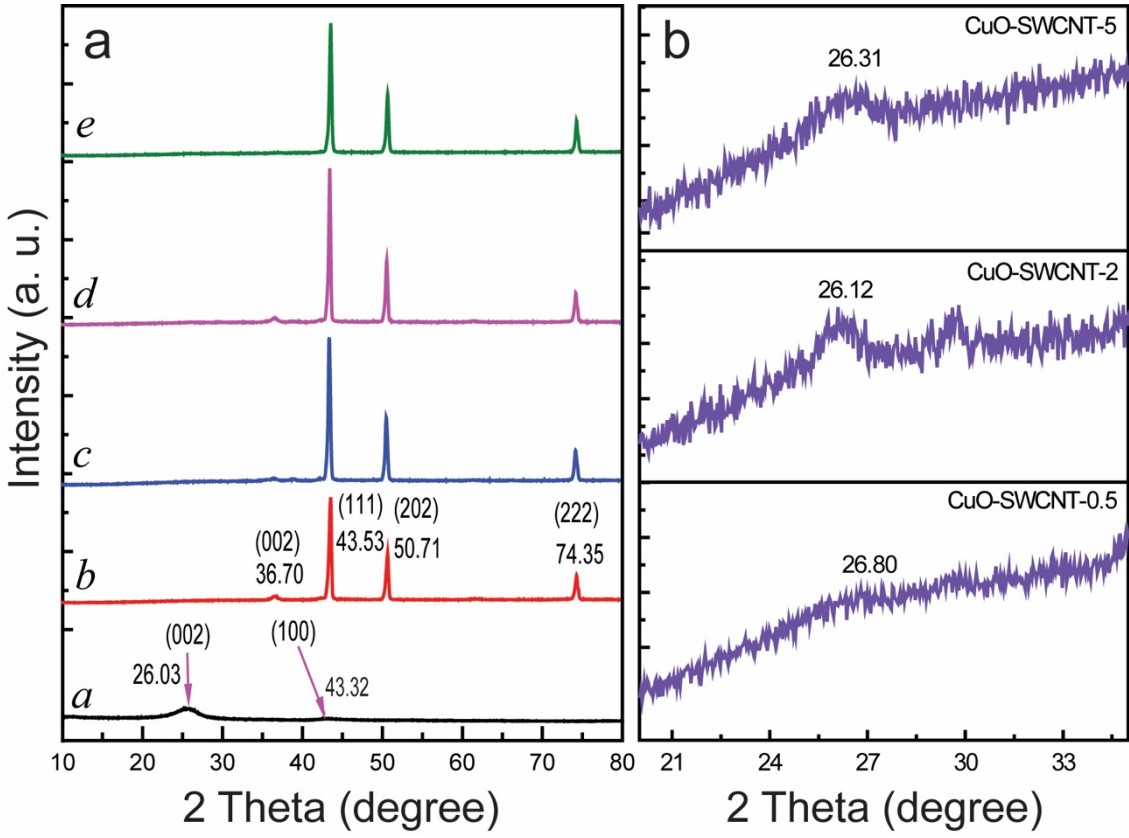

**Figure 3.** XRD patterns of the (**a**) (curve *a*) pristine SWCNTs, (curve *b*) pristine CuO, (curve *c*) CuO-SWCNT-0.5 nanocomposite, (curve *d*) CuO-SWCNT-2 nanocomposite, and (curve *e*) the CuO-SWCNT-5 nanocomposite; (**b**) expanded region to locate carbon peaks in the XRD pattern of the nanocomposites.

The XRD pattern of CuO was also used to determine its crystallite size. The size of the crystallites was computed by the Scherrer equation (Equation (1))

$$D = \frac{k\,\lambda}{\beta\,\cos\theta} \tag{1}$$

where D is the average diameter of the CuO crystallites, $\lambda$ is the wavelength of the X-rays used ($\lambda$ = 0.15406 nm for Cu K$\alpha$ radiation), $k$ is Scherrer's constant ($k$ = 0.90), $\theta$ is the diffraction angle, and $\beta$ is the angular line width (in radians) at the half-maximum intensity (i.e., the full-width at half-maximum (FWHM)). The peak that emerged at 43.53° (2$\theta$) due to the reflection from the (111) plane (the most prominent peak) was used to assess the crystallites' size, which was determined to be 24.82 nm.

Curves c to e in Figure 3a depict the XRD patterns of the CuO-SWCNT nanocomposites prepared at different calcination times, i.e., CuO-SWCNT-0.5, CuO-SWCNT-2, and CuO-SWCNT-5, respectively. The peaks at 36.70°, 43.53°, 50.71°, and 74.35° observed in the patterns indicate that the monoclinic crystalline form of the CuO is not altered in the CuO-SWCNT composites. The carbon portion in the spectrum of CuO-SWCNT nanocomposites has been magnified in Figure 3b, which displays the peaks at 26° (with slight differences in decimal values). The other peak at 43.32° could not be distinguished in the patterns of the composites because it is superimposed with the most intense CuO peak. The (002) carbon peak appears to be less distinct in the patterns of the composites because of its comparatively low intensity. The almost-unchanged positions of the carbon peaks suggest that the crystallinity of carbon remains intact in the CuO-SWCNT nanocomposites.

The chemical constituents with their oxidation states in the synthesized nanocomposites were analyzed by high-performance XPS; the results are presented in Figures S1 and S2, Figure 4. The high-resolution survey spectrum of the CuO-SWCNT-5 nanocomposite (Figure 4a) confirms that the nanocomposite comprises three elements: Cu, O, and C (their oxidation states are elaborated later in this section). The Cu-2*p* core-level spectra of the CuO-SWCNT-5 nanocomposite are presented in Figure 4b. The broad and asymmetrical range has been deconvoluted, revealing two discrete parts, indicating the coexistence of two forms of $Cu^{2+}$ ions in nonequivalent chemical environments. The two maxima at 933.30 and 953.32 eV correspond to the $Cu^{2+}$ state in the CuO-SWCNT nanocomposites. The binding energy difference between these two peaks, as calculated from the spectra, is 20.02 eV, which matches the value reported for $Cu^{2+}$ in the literature. In addition to these peaks, two characteristic satellite peaks are positioned at 943.83 and 962.65 eV in the spectrum of pure CuO, indicating that the copper in pure CuO is $Cu^{2+}$ [20,21].

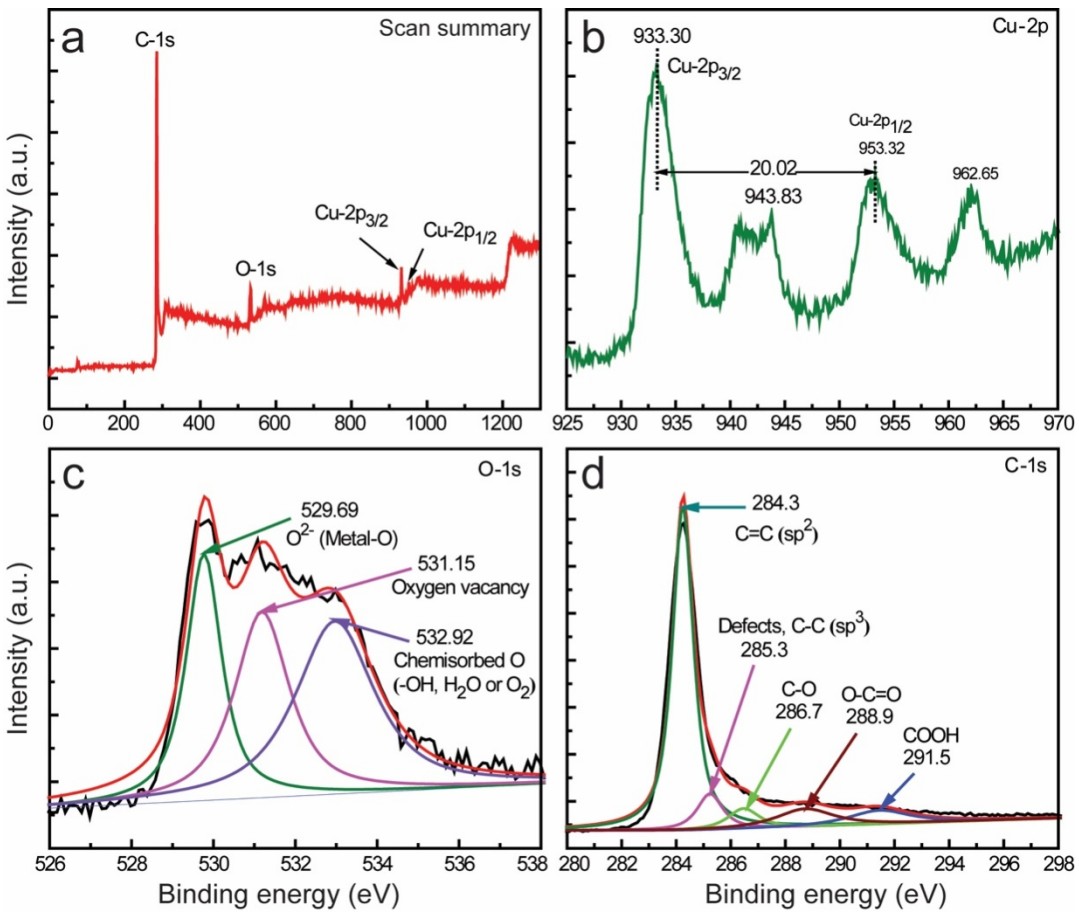

**Figure 4.** XPS spectra of the composites: (**a**) survey spectra of the synthesized CuO-SWCNT-5 nanocomposite, and high-resolution spectra displaying (**b**) Cu-2*p* scan demonstrating Cu-2$p_{3/2}$ and Cu-2$p_{1/2}$ peaks, (**c**) the core level of O 1*s*, and (**d**) the core level of C 1*s* in the CuO-SWCNT-5 nanocomposite.

Figure 4c displays the O-1*s* core-level spectra of CuO-SWCNT-5 nanocomposite. The extensive asymmetry of the spectra led us to deconvolute them into three separate curves, indicating the presence of different chemical environments adjacent to the oxide ions. The relatively stronger section of the O-1*s* spectrum at the lower binding energy (529.69 eV) indicates that $O^{2-}$ ions are bonded to the $Cu^{2+}$ ions of monoclinic CuO. The other component at 531.15 eV represents the vacant oxygen sites. The third component at a higher binding energy represents surface-chemisorbed O in the form of OH,

$H_2O$, or $O_2$, indicating the existence of loosely bound oxygen species adjoining the surface region of the CuO nanocrystals [9].

The C-1*s* core-level profiles of CuO-SWCNT-5 composite are displayed in Figure 4d. The comparatively robust peak at 284.3 eV indicates that the SWCNT sample mostly contains $sp^2$-hybridized graphitic carbon. However, the peak at 285.3 indicates the presence of some defects in the form of C–C carbon. The other peaks at 286.7, 288.9, and 291.5 eV are ascribed to C–O, O–C=O, and COOH groups, correspondingly present in the nanocomposite [1,9]. The bonding states of the constituent elements, as mentioned above, assert that CuO nanocrystals form heterojunctions with SWCNTs through the formation of covalent bonds—specifically, CuO-SWCNTs or Cu–OOC–SWCNTs—or through bonding via van der Waals forces.

The XPS spectra of the two other samples, (i.e., CuO-SWCNT-2 and CuO-SWCNT-0.5) which are mostly consistent with those of the CuO-SWCNT-5 are presented in Supplementary Information (Figures S1 and S2). In Figure S1a–c, the high resolution XPS spectra of Cu-2p$_{3/2}$ and Cu-2p$_{1/2}$ peaks, O-1s core level and C-1s core level spectra of CuO-SWCNT-0.5 nanocomposite are presented. Similarly, Figure S2 displays the corresponding XPS spectra of CuO-SWCNT-2 samples. The Cu-2p$_{3/2}$ and Cu-2p$_{1/2}$ spectra in both Figures S1 and S2 demonstrate that the divalent oxidation state of Cu remains intact in both composites [20].

### 2.3. Thermal Properties of the Nanocomposites

The thermal stability of the synthesized CuO-SWCNT nanocomposites was investigated using a thermal analyzer. Nanocomposite samples were subjected to heat treatment between room temperature (25 °C) and 800 °C under an $N_2$ atmosphere; the heating rate was 10 °C min$^{-1}$. The corresponding results are shown in Figure 5. The curves for SWCNTs are presented separately because of the large differences in the weight-loss percentages.

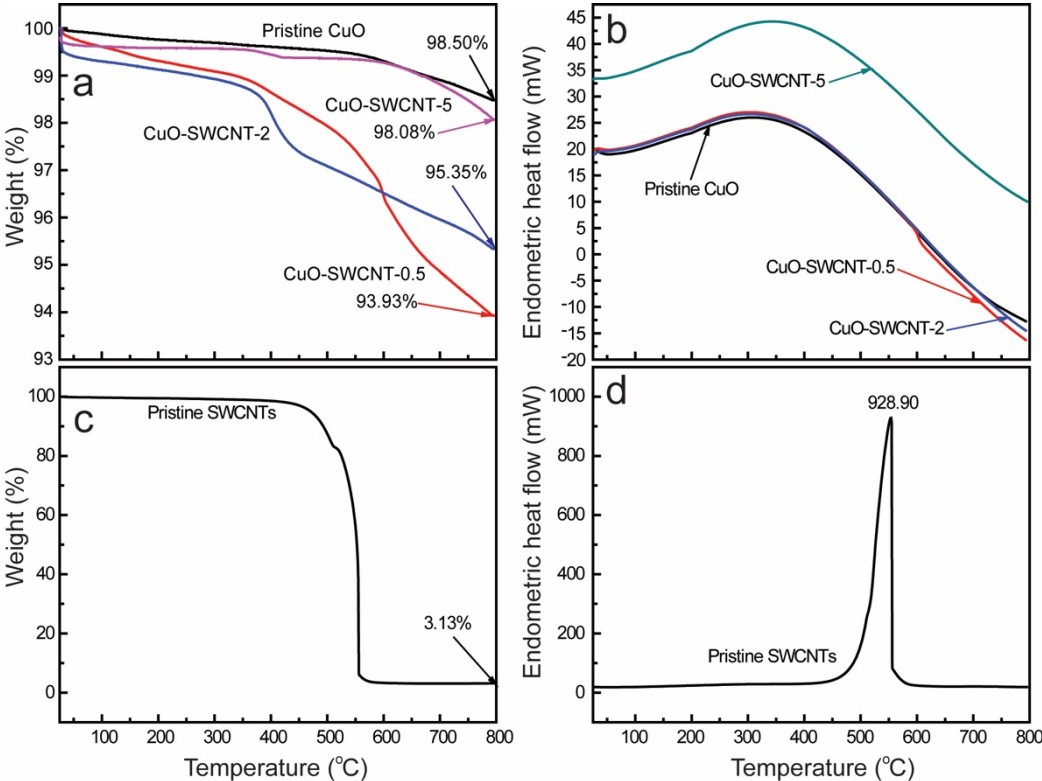

**Figure 5.** Thermal analysis of CuO–SWCNT nanocomposites and the pristine CuO: (**a**) Thermogravimetric analysis (TGA) plot and (**b**) Differential scanning calorimetry (DSC) plot. Thermal analysis of the pristine SWCNTs: (**c**) TGA plot and (**d**) DSC plot.

A slight mass loss is observed near 100 °C for pristine CuO (Figure 5a), which is attributed to the vaporization of physisorbed water molecules. The continuous and gradual loss of mass between 550 to 780 °C is ascribed to the decomposition of remnant acetates [28]. No additional decomposition is detected at temperatures as high as 800 °C, and the residual mass of the pristine CuO is 98.50%.

The slight weight loss of the pristine SWCNTs (Figure 5c) between 100 and 400 °C is assigned to the dehydration of adsorbed moisture and to the degradation of some hydroxyl groups contained in the SWCNTs. The rapid weight loss at temperatures greater than 450 °C is attributed to the deterioration of the SWCNTs [28]. Nearly 97% of the carbon content was decomposed as the temperature was increased to 560 °C. However, the CuO-SWCNT nanocomposites appear to be very stable (thermally) compared to the CNTs. No substantial degradation of the nanocomposites is observed at temperatures of 500 °C or less. When the temperature is increased to 800 °C, the decomposition of the nanocomposites is nearly negligible compared with that of the pristine SWCNTs but appears to be comparable to that of the pristine CuO. The CuO-SWCNT-5 is almost as stable as the pristine CuO, which is attributable to the complete elimination of water adsorbates, surface hydroxyl groups, and acetate groups during the 5 h calcination (5 h was the longest calcination time) [14,28]. These observations affirm that the fabricated nanocomposites are thermally very stable compared with the SWCNTs. The increased thermal stability indicates the formation of strong chemical bonds between CuO and SWCNTs [11]. The TGA results also validate the intimate association of CuO nanoparticles on the SWCNT surfaces, which is consistent with the results of the FE-SEM, HR-TEM, and XPS analyses. The DSC curves demonstrate that all the thermal operations were endothermic in nature [28].

## 2.4. Optical Properties of the Nanocomposites

The improvement in the bandgap energy of the as-synthesized nanocomposites was accessed through the Tauc Plot method. The UV-vis absorbance spectra of the composites and pristine CuO were recorded from their suspensions dispersed in ethanol in very low concentrations (0.005 mg mL$^{-1}$). Separate Tauc Plots were obtained for different samples and are presented in Figure S3. The figures display that the bandgap energy of pristine CuO is 1.71 eV, and is reduced successively to have values of 1.56 eV in CuO-SWCNT-0.5, 1.54 eV in CuO-SWCNT-2 and 1.50 eV in CuO-SWCNT-5. It is evident from the plots that there is a significant improvement in the bandgap energy of CuO due to the formation of heterojunction with SWCNTs. Such an improvement in bandgap energy leads to the production of substantial number of electron–hole pairs in the semiconductor catalyst so that they participate in the photocatalytic degradation of organic dye [1]. Among the as-synthesized nanocomposites, CuO-SWCNT-5 holds the least bandgap energy value.

## 2.5. Specific Surface Area and Pore Volume Studies

The specific surface area, pore volume and pore size of the fabricated nanocomposites were analyzed via Brunauer-Emmett-Teller (BET)/Barrett-Joyner-Halenda (BJH) measurements and the corresponding values are summarized in Table 1. It is discernible that the specific surface area of the CuO nanoparticles increased when increasing the calcination time (from CuO-SWCNT-0.5 to CuO-SWCNT-5). These results assert that the increased surface area of CuO can be attained by the formation of heterojunction with SWCNTs [20]. The BET surface area results reveal that the CuO-SWCNT-5 nanocomposite possesses the highest specific surface area value among the as-synthesized nanocomposites.

**Table 1.** Experimentally determined values of specific surface area ($S_{BET}$), total pore volume ($V_{pore}$), and average pore diameter ($D_{pore}$).

| Catalyst | Specific Surface Area ($S_{BET}$) ($m^2\ g^{-1}$) | Total Pore Volume ($V_{pore}$) ($cm^3\ g^{-1}$) | Average Pore Diameter ($D_{pore}$) (nm) |
|---|---|---|---|
| CuO | 33.72 | 0.02 | 3.32 |
| CuO-SWCNT-0.5 | 34.08 | 0.10 | 12.89 |
| CuO-SWCNT-2 | 35.64 | 0.03 | 3.73 |
| CuO-SWCNT-5 | 40.82 | 0.05 | 4.84 |
| SWCNTs | 256.83 | 0.51 | 8.10 |

## 2.6. Photocatalytic Action

The photocatalytic performances of the pristine CuO nanocrystals and CuO-SWCNT nanocomposites fabricated using the same method under identical conditions were examined by assessing the degradation of MB under natural sunlight exposure. Figure 6 depicts the photocatalytic abilities of all the samples under identical conditions. The absorbance spectra corresponding to the photodegradation of MB in the UV-vis range from 200 to 900 nm are displayed in Figures S4 and S5. Exposure to solar light can cause self-degradation of MB; therefore, a blank test was performed under solar-light irradiation alone without the use of a catalyst. The blank test (Figure S6) showed negligible self-deterioration of the MB solution under solar light.

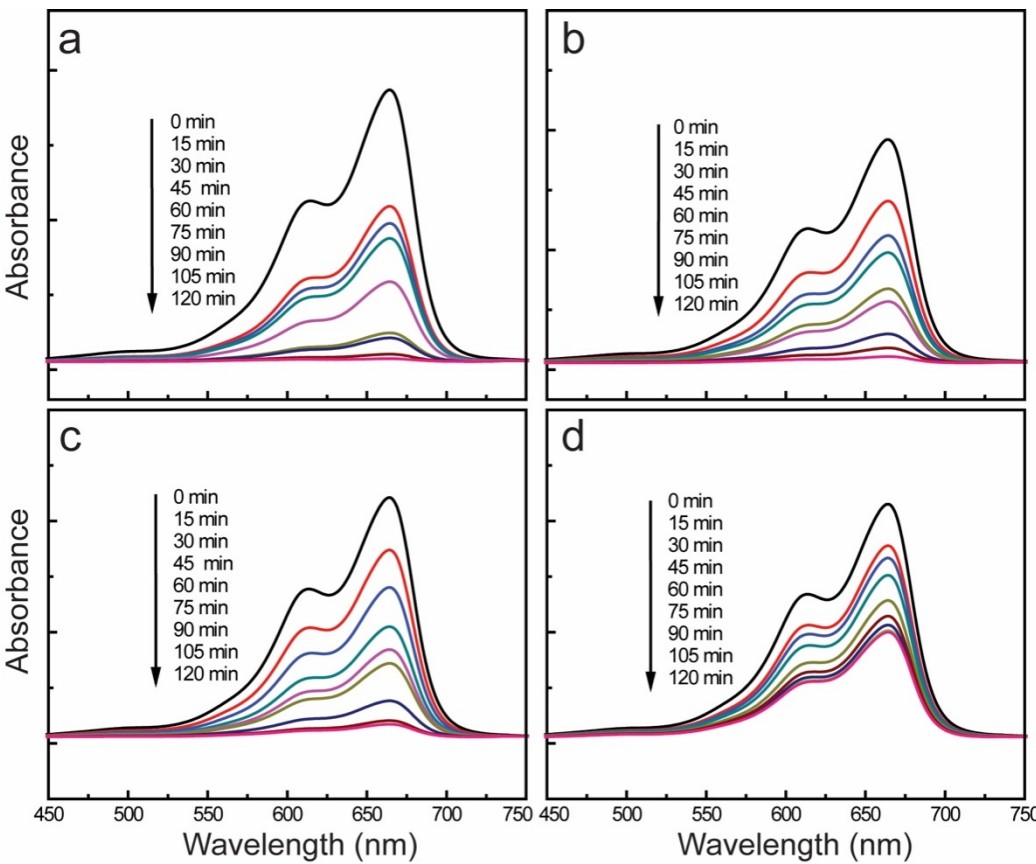

**Figure 6.** UV-vis absorbance spectra of the methylene blue (MB) solution as a function of time, showing the photodegradation achieved with (**a**) CuO-SWCNT-5, (**b**) CuO-SWCNT-2, (**c**) CuO-SWCNT-0.5, and (**d**) pristine CuO nanocrystals.

Figure 6 shows that the CuO-SWCNT-5 photocatalyst exhibits the highest photocatalytic ability among the investigated samples, i.e., it caused 97.33% photodegradation of MB during 2 h of solar

irradiation. The CuO-SWCNT-2 and CuO-SWCNT-0.5 achieved 94.24% and 93.15% deterioration, respectively, within the same time period. For comparison, the pristine CuO achieved 54.20% degradation of MB within the same time period and under the same irradiation conditions. As shown in the figures, we achieved a substantial improvement in the photocatalytic efficiency of CuO by combining it with SWCNTs. The enhanced photocatalytic efficiency of the CuO-SWCNT nanocomposites may be because of the synergistic effects the CuO and SWCNTs.

The photodeterioration of MB by the action of the as-fabricated catalysts followed a pseudo-first-order kinetic model (Figure 7a). The degradation reaction rate constant was computed by the following equation

$$\ln\left(\frac{C_o}{C}\right) = kt \tag{2}$$

where $C_o$ is the initial concentration, $C$ is the concentration at the assessed time, $k$ is the photodegradation reaction rate constant ($min^{-1}$), and $t$ is the time interval of irradiation (min). The calculated values for the degradation rate constant were 0.0252, 0.0190, 0.0178, and 0.0084 $min^{-1}$ for CuO-SWCNT-5, CuO-SWCNT-2, CuO-SWCNT-0.5, and pristine CuO nanocrystals, respectively. The reusability of the CuO-SWCNT-5 nanocomposite was evaluated by using it as photocatalyst for three successive cycles under similar reaction conditions. As demonstrated by Figure 7b, the photocatalytic efficiency of the CuO-SWCNT-5 nanocomposite remains unaltered when the catalyst is recycled.

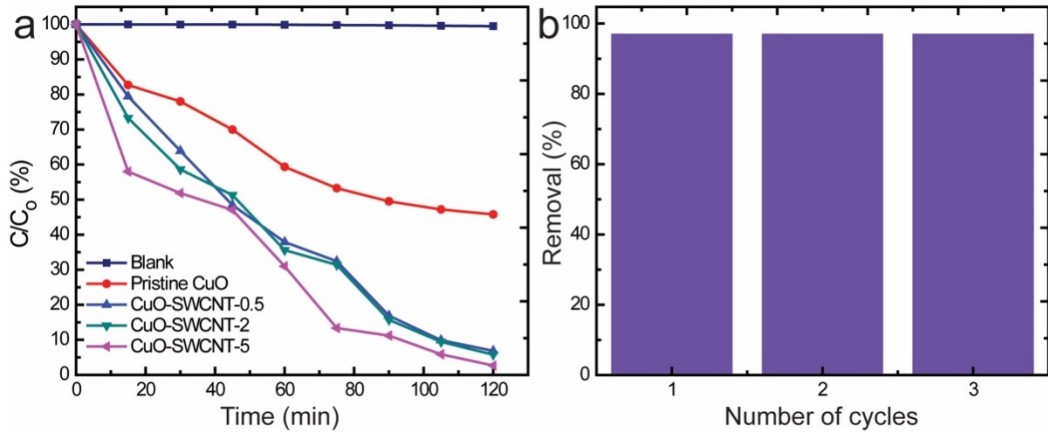

**Figure 7.** (**a**) Comparison of the degradation of MB by the photocatalytic action of different samples under solar irradiation; (**b**) recycling performance of the CuO-SWCNT-5 photocatalyst.

The UV-vis absorbance spectra (Figure 6) and degradation rate constant values (Figure 7a) demonstrate that CuO-SWCNT-5 possesses the highest photocatalytic efficiency in comparison to other samples and the control. In comparison to the photocatalytic efficiency of pristine CuO (i.e., 54.20% degradation of MB), the CuO-SWCNT-5 is 43.13% more effective, whereas CuO-SWCNT-2 is 40.04% more effective and CuO-SWCNT-0.5 is 38.95% more effective. The relatively higher photocatalytic performance of the CuO-SWCNT-5 catalyst is attributable to the complete formation of heterojunctions between CuO and the carbon surface during the longest heat treatment (5 h). It is also favored by the noteworthy improvement in the bandgap energy and higher BET surface area values.

The photocatalytic performance of our nanocomposite was compared with that of the previously reported composites comprised of CuO as the main constituent. The comparative results are revealed in Table 2. The results show how our photocatalyst is more effective compared to the others available in the literature.

**Table 2.** Comparison of photocatalytic performance of our photocatalyst with other CuO based catalysts.

| Composite | Pollutant | Pollutant Concentration | Composite Doze | Degradation (%) | Degradation Time (min) | Ref. |
|---|---|---|---|---|---|---|
| CuO-CNT | DR31 and RR120 | 50 mg L$^{-1}$ | 0.005g/800mL | 89 87 | 180 | [26] |
| CuO-CNT | PCA | 10 mg L$^{-1}$ | 0.375 g L$^{-1}$ | 97 | 180 | [25] |
| CuO-ZnO | Phenol | 10 mg L$^{-1}$ | 50mg/100 mL | 78 | 180 | [20] |
| CuO-CuWO$_4$ | Phenol X3B dye | 0.22 mM 0.066 mM | 1.7 g L$^{-1}$ | N/A | 300 120 | [23] |
| CuO-Cu$_2$O-Cu | RhB | $2.5 \times 10^{-5}$ mol L$^{-1}$ | N/A | N/A | 120 | [3] |
| CuO-SWCNT | MB | 0.10 mg mL$^{-1}$ | 150mg/100 mL | 97.33 | 120 | Our work |

## 2.7. Proposed Mechanism of Photocatalysis

When a CuO-SWCNT photocatalyst in MB solution is exposed to sunlight with energy ($h\nu$) equal to or greater than its bandgap energy, the electrons in the occupied valence band (VB) of the photocatalyst are excited and migrate to the unoccupied conduction band (CB). Consequently, the CB and VB contain photogenerated electrons ($e^-$) and positively charged holes ($h^+$), respectively [26]. Because CuO possesses a lower VB edge potential than the SWCNTs, the electrons present in the CB of the SWCNTs instantly migrate to the CuO segment through the heterojunctions. In the meantime, the photon-induced holes in the VB of CuO freely migrate to the SWCNTs along the heterojunctions, because the VB edge potential of CuO is exceedingly positive compared with the lowest unoccupied molecular orbital (LUMO) of the SWCNTs [25]. Consequently, the recombination probability for the photon-induced electron–hole pairs is greatly reduced, and the surplus electrons and holes are available to trigger the redox reactions. The synergistic effects initiated by both CuO and SWCNTs greatly enhance the photooxidation of MB. The proposed mechanism of photocatalytic degradation of the MB solution is displayed in Figure 8.

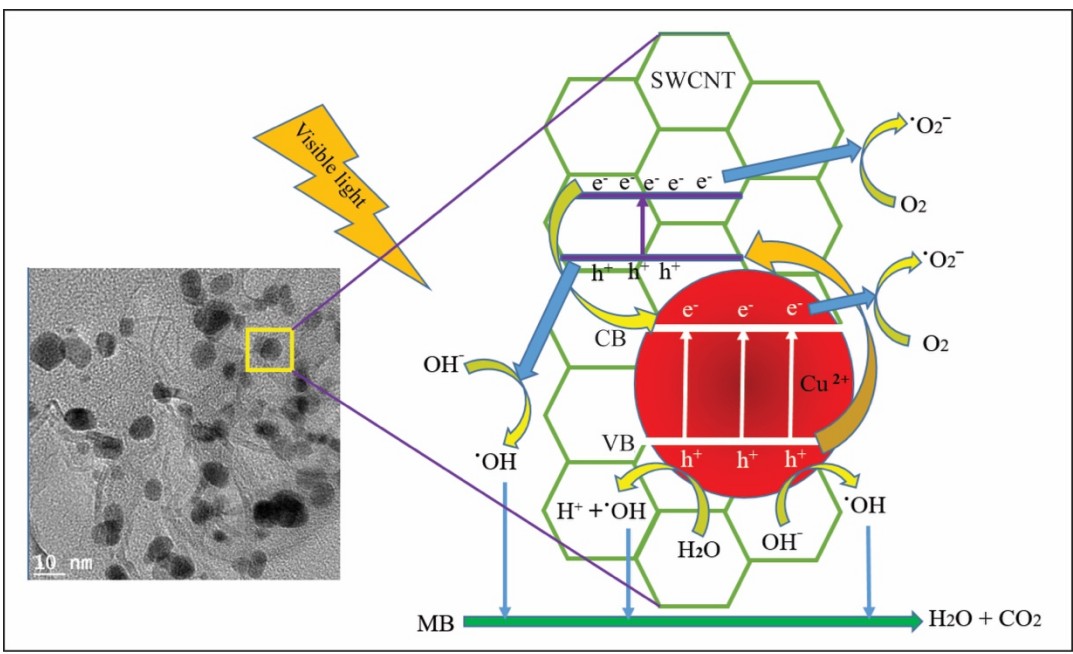

**Figure 8.** Proposed mechanism of generation of active species and deterioration of MB in the presence of sunlight (valence band (VB) and conduction band (CB) levels as well as particle sizes are not drawn to scale).

Further, the electrons and holes are made available for redox reactions by oxygen vacancy and surface defects. The presence of oxygen vacancy ($V_o$) defects in CuO and surface defects (SD) between

the VB and CB of the SWCNTs (as revealed by XPS) also promote photodegradation under visible light. The migration of photoexcited electrons from the VB to the SD and $V_o$ sites, and from these regions to the CB, leads to the generation of additional electron–hole pairs [25]. The photogenerated electrons successively reduce $O_2$ to generate superoxide radicals ($\cdot O_2{}^-$), which are promptly reduced to powerful oxidants, i.e., hydroxyl radicals ($\cdot OH$) [20]. These hydroxyl radicals degrade MB molecules, which are adsorbed onto the surface of the photocatalyst.

$$CuO\text{-}SWCNT + h\nu \quad \rightarrow \quad e^-{}_{(CB)} + h^+{}_{(VB)} \tag{3}$$

$$e^-{}_{(CB)} + O_2 \quad \rightarrow \quad \cdot O_2{}^- \tag{4}$$

Holes attack water molecules to produce hydroxyl radicals and hydrogen ions ($H^+$) and initiate other reactions (Equations (5)–(11))

$$h^+{}_{(VB)} + H_2O \quad \rightarrow \quad H^+ + \cdot OH \tag{5}$$

$$\cdot O_2{}^- + H^+ \quad \rightarrow \quad \cdot HO_2 \tag{6}$$

$$\cdot HO_2 + \cdot HO_2 \quad \rightarrow \quad O_2 + H_2O_2 \tag{7}$$

$$\cdot HO_2 + H^+ + e^- \quad \rightarrow \quad H_2O_2 \tag{8}$$

$$H_2O_2 + e^- \quad \rightarrow \quad OH^- + \cdot OH \tag{9}$$

$$H_2O_2 + h\nu \quad \rightarrow \quad 2\,\cdot OH \tag{10}$$

$$\cdot OH + MB \quad \rightarrow \quad \cdot R \text{ (intermediates)} \rightarrow CO_2 + H_2O \tag{11}$$

The holes also react with surface-absorbed hydroxyl groups ($OH^-$) present in water to generate $\cdot OH$ radicals, which attack MB molecules (Equation (12)). The holes can invade the MB molecules directly and trigger their oxidation (Equation (13))

$$h^+ + OH^- \quad \rightarrow \quad \cdot OH \tag{12}$$

$$h^+ + MB \quad \rightarrow \cdot R \text{ (Intermediates)} \rightarrow CO_2 + H_2O \tag{13}$$

The $\cdot OH$ radicals are the main oxidizing agents that attack MB molecules instantly and create transitional compounds or intermediates. Those intermediates are further invaded by $\cdot OH$ and are transformed into harmless inorganic products, $H_2O$ and $CO_2$. We point out that this mechanism is not yet directly proven and may be further tested by electron paramagnetic resonance spectroscopy and intermediate recognition.

## 3. Experimental Section

### 3.1. Chemicals

Copper(II) acetate hydrate (Sigma-Aldrich, 98% purity, St. Louis, MO, USA), ethanol (Sigma-Aldrich, 99.5% purity), SWCNTs with an outer diameter of 1–2 nm (>90% purity, US Research Nanomaterials, Inc., Houston, TX, USA), and MB (Alfa Aesar, high purity, Heysham, Lancashire, UK) were used without further purification. MB solutions with the requisite concentrations were prepared with distilled water.

### 3.2. Preparation of CuO-SWCNT Nanocomposites

In a particular experiment, copper(II) acetate hydrate (6.00 g) was mixed with ethanol (100 mL) in a graduated beaker, and a homogeneous solution was prepared by subjecting the mixture to bath sonication at room temperature (22 °C) for 1 h. SWCNTs (100 mg) were added to this prepared

solution under magnetic stirring (1 h). The mixture was left undisturbed for recrystallization for 6 h after the magnetic stirrer was removed. Crystals appeared immediately after the stirrer was removed; however, ~6 h was needed for the complete and sustained recrystallization of copper(II) acetate throughout the SWCNTs. The copper(II) acetate crystal-SWCNT mixture was separated from the ethanol solvent by vacuum filtration. The residue was collected and dried for 2 h in a furnace at 60 °C to vaporize the ethanol completely. The product was then heated in a muffle-type furnace (KSL-1100X-S-UL-LD). For this purpose, the dried product was placed in a quartz crucible with a cover and the crucible was subsequently inserted into a vacuum chamber. The chamber was sealed using an oxygen-free, copper ring gasket (SUS 314), and the sample was calcined at 500 °C to synthesize the CuO-SWCNT nanocomposites. We prepared different CuO-SWCNT nanocomposite samples by varying the calcination time (at the same temperature, i.e., 500 °C). The nanocomposites prepared with calcination times of 30 min, 2 h, and 5 h were named as CuO-SWCNT-0.5, CuO-SWCNT-2, and CuO-SWCNT-5, respectively. In all samples, a fixed weight of SWCNTs (i.e., 0.100g) was used, whereas the average CuO contents in each of the CuO-SWCNT-0.5, CuO-SWCNT-2, and CuO-SWCNT-5 were 1.6880, 1.6812 and 1.6716 g, respectively. The corresponding molar ratios of CuO:SWCNTs were 2.547:1, 2.536:1 and 2.531:1, respectively. Thus, the average CuO-to-SWCNT molar ratio in all of the prepared CuO-SWCNT composites was 2.5:1. The proportions were worked out from the weights of SWCNTs and CuO-SWCNT assemblies by repeating the experiments three times.

### 3.3. Characterization

Morphological characteristics of the fabricated nanocomposites were examined by field-emission scanning electron microscopy (FE-SEM, HITACHI, SUB 8230, Hitachi High-Tech Corporation, Minato-ku, Tokyo, Japan) and high-resolution transmission electron microscopy (HR-TEM, JEM -2200FS, Jeol Ltd. Akishima, Tokyo, Japan). The structural characteristics were assessed through high-resolution X-ray diffraction (XRD, Smart Lab, Rigaku, MA, USA). Spectroscopic characteristics were examined through X-ray photoelectron spectroscopy (XPS, K-Alpha, Thermo Scientific, Waltham, USA). The thermal stability of the nanocomposites was evaluated using a thermal analyzer (SDT Q600 V20.9 Build 20, TA Instruments, DE, USA). The thermal analysis of pristine CuO and the CuO-SWCNT nanocomposites was performed in $N_2$ atmosphere at a heating rate of 10 °C $min^{-1}$, while that of SWCNTs was performed in air at the same heating rate. BET/BJH analysis was performed via Accelerated Surface Area and Porosimetry system (ASAP 2420 V2.09 (V2.09 I, micromeritics, Georgia, USA)).

### 3.4. Fabrication of Pristine CuO Nanocrystals

Pristine CuO nanocrystals were synthesized using copper(II) acetate monohydrate and ethanol according the procedure elaborated in Section 2.2. The thus-fabricated CuO nanocrystals were used in control experiments for elucidating the photocatalytic efficiencies of the CuO-SWCNT nanocomposites.

### 3.5. Photocatalytic Experiments

In each set of experiments, a CuO-SWCNT photocatalyst (150 mg) was mixed with 100 mL of MB solution (0.10 mg $mL^{-1}$) prepared with distilled water. The mixture was bath-sonicated for 1 h to ensure homogeneous mixing. It was left undisturbed in a dark chamber (1 h) to attain adsorption–desorption equilibrium between the MB molecules and the active sites of the photocatalyst. The photocatalytic experiments were carried out on a sunny day in an outdoor environment under uninterrupted sunlight between 12:30 and 2:00 p.m., when the fluctuations in solar intensity are minimal. The outside temperature was between 23 and 25 °C, and the average solar irradiance (radiation flux per unit area) was ~950 W $m^{-2}$. The intensity of the solar irradiance was measured with a solar power meter TM-206 (TENMARS ELECTRONICS, Taipei, Taiwan). The photocatalytic activities were evaluated through the degradation of MB while ensuring the use of the visible portion of sunlight. UV rays were excluded from the vessel by a UV cutoff filter that allowed only visible and near-infrared radiation to

influence the photoreaction. Photodegradation of the MB solution was evaluated through absorbance measurements using a UV-vis spectrophotometer.

*3.6. Reusability Test*

To investigate the recyclability of the CuO-SWCNT photocatalysts, each used CuO-SWCNT nanocomposite was collected, washed with distilled water, and dried at 100 °C for ~15 min in preparation for its reuse. The photocatalysts were recycled three times. The photocatalytic experiments were conducted between 12:30 to 2:00 p.m. for three days at the same location to minimize fluctuations in environmental conditions.

## 4. Conclusions

CuO-SWCNT nanocomposites with good photocatalytic performance under natural sunlight were synthesized via cost-effective, facile recrystallization followed by calcination. The nanocomposites contained SWCNTs surrounded by CuO nanocrystals through direct chemical bonds. XPS, XRD, TEM, and TGA/DSC results confirm the formation of heterojunctions between the CuO and SWCNT surfaces. The photocatalytic capabilities of the composites were assessed by evaluating the photodegradation of MB as a function of time. All samples showed remarkable photocatalytic performance; however, the CuO-SWCNT-5 photocatalyst demonstrated the best performance of 97.33% MB decomposition in 2 h. The formation of heterojunctions between the CuO and SWCNT increases the separation and escalates the transfer of photoinduced electron–hole pairs. It also enhances visible-light absorption and results in remarkable photocatalytic efficiency. The synthesized nanocomposites can efficiently photocatalyze the deterioration of persistent pollutants such as MB into harmless compounds in the presence of sunlight.

**Supplementary Materials:** The following are available online at http://www.mdpi.com/2073-4344/10/3/297/s1, Figure S1: High resolution XPS spectra of (a) Cu-$2p_{3/2}$ and Cu-$2p_{1/2}$ peaks, (b) O-1s core level and (c) C-1s core level of CuO-SWCNT-0.5 nanocomposite, Figure S2: High resolution XPS spectra of (a) Cu-$2p_{3/2}$ and Cu-$2p_{1/2}$ peaks, (b) O-1s core level and (c) C-1s core level of CuO-SWCNT-2 nanocomposite, Figure S3: Tauc Plots of CuO-SWCNT nanocomposites and pristine CuO depicting the reduction in the bandgap of CuO due to the formation of heterojunction with SWCNTs, Figure S4: UV-vis absorbance spectra showing the photodegradation of MB by the action of the CuO-SWCNT-5 photocatalyst (in the full absorption range from 200 to 900 nm), Figure S5: UV-vis absorbance spectra of photodegradation of MB by the action of photocatalysts: (a) CuO-SWCNT-2 and (b) CuO-SWCNT-0.5 (in the full absorption range from 200 to 900 nm), and Figure S6: Blank test; effect of solar light alone on the decomposition of MB in the absence of a catalyst.

**Author Contributions:** J.R.H. supervised and provided guidance in entire research work, K.P.S. designed and executed experiments, analyzed data and wrote the article, I.L. made provisions of materials, analysis equipment and reagents, I.L., M.A.I., M.A.H., and J.A. analyzed data and read the manuscript rigorously to upgrade it. All authors have read and agreed to the published version of the manuscript.

**Funding:** This research has been supported by the Korean Government (NRF grants 2018R1A2B6006155).

**Conflicts of Interest:** The authors assert no conflicting interests.

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
