# Peer review of "Enhanced Visible-Light Photocatalysis of Nanocomposites of Copper Oxide and Single-Walled Carbon Nanotubes for the Degradation of Methylene Blue"

_catalysts, doi:10.3390/catal10030297_

Round 1

Reviewer 1 Report

The manuscript is focused on the production of CuO–SWCNT nanocompites and on the evaluation of their photocatalytic performances under visible-light excitation. The work plan is scientifically correct, but, in my opinion some additional experimental details and material characterizations should be provided. Overall, some aspects need to be improved and the manuscript is not suitable for publication in this form. Major revisions are needed.

More in details the following aspects need improvement and the following questions/comments are expected to be discussed in the text.

Entire manuscript

Even if the manuscript is nicely written and it is easy to follow a linguistic check should be performed (check verbs and plurals).

Abstract

Line 19: I suggest to change the word thermolysis with calcination.

Introduction:

The authors should highlight better the novelty proposed by the manuscript underlying the difference between the proposed study and those already available in the pertinent literature on CuO-SWCNTs materials

Results and discussion section:

2 is blurred, please improve the readability 3: lines a and b are too close and it is not easy to read the peaks, please improve the readability distributing better the curves along the vertical. 4: the XPS spectra are related to only one nanocomposite, please specify which one is and please provide those of the other samples as supplementary material. The interpretation of the thermogravimetric data should be improved. In my opinion, since the materials will be used in photocatalytic applications at temperature below 100°C the evaluation of the thermal stability of the materials up to 800°C is not so relevant. For this reason I suggest to use the TGA data for other kind of considerations. The TG curves (I suppose obtained by analyses under oxidative atmosphere (air)) can be used to obtain information about the actual content of CuO inside the composites and to highlight differences among the three samples due to the calcination duration time (CuO–SWCNT0.5 exhibited the highest weight loss compared to the other two materials indicating that a complete conversion to CuO was not achieved in 0.5 h). Moreover it is obvious that the CuO–SWCNT nanocomposites are more stable that pure SWCNTs, being the inorganic component the most abundant and not reactive under oxidative atmosphere! Information about the actual CuO content in each nanocomposite should be provided. Information about the specific surface area and pore size distribution of the different materials should be provided since these data are very helpful for the interpretation of the catalytic performances of the materials. Details about the actual band-gaps of the nanocomposites should be provided for a better interpretation of the catalytic performances of the materials and to shed light on the heterojunction between the two nanocomposite components. Lines 248-252: quantify the improvement in the photocatalytic activity passing from bare CuO to the nanocomposites. A comparison between the performances of the proposed materials and those of CuO containing materials reported in the pertinent literature can strength the relevance of the proposed results. Some considerations about the reasons why CuO–SWCNT5 is the best performing material should be provided. The authors provide some hypotheses about the mechanism, but they do not provide experimental proofs to support them (EPR measurements, intermediates recognition, etc).

Experimental section:

A determination of the actual Cu content should be provided as well the actual ratio between CuO and SWCNTs in each nanocomposite. The experimental conditions used during TG analysis should be reported. Please add details about the XPS data elaboration Please specify how the intensity of solar irradiance was measured Did the authors performed leaching tests to evaluate loss of CuO or unreacted species (acetate ions, Cu2+ ions) during photocatalytic tests?

Reviewer 2 Report

This manuscript reports an interesting work about the preparation and characterization of Copper Oxide and Single-Walled Carbon Nanotubes Nanocomposites and their application as catalysts for the degradation of MB. There is a good characterization of the synthesized materials by using different techniques such as SEM, TEM, XRD, XPS, and TG. However, I think that in this case, the authors should also supply the surface area and pore size of materials based on BET studies.

In addition, it would also be interesting to see some characterization of the catalytic material after the oxidation reactions. Did a leaching process take place? Was the reaction mixture analyzed after the first catalytic cycle?

I also think that the title of the article is very general considering that the authors only used a single substrate in their catalytic studies. The authors should test more substrates to confirm the effect they indicate on the title of the manuscript.

Reviewer 3 Report

Authors reported here synthesis of CuO decorated-SWCNT and used them for MB decomposition under light. Though the authors tried to characterize the material and applied them for photocatalytic dye degradation but still there is no novelty in this study. Thousands of paper on this same topic is available. Therefore this work is just looks like a repetition of previous published works. Therefore I strongly recommend not to publish this work in Catalysts. I also suggest authors to explore other photocatalytic application to improve the quality of the paper. 

Round 2

Reviewer 1 Report

The authors addressed most of the comments and questions I formulated during the first round of revision. The revised version of the manuscript is more complete and readable. In this form the manuscript is suitable for publication to me.

Reviewer 2 Report

The authors added the BET analysis data through revision and improved the manuscript in general. I have no additional comments concerning this manuscript.

Reviewer 3 Report

I recommend to accept it in present form